# A Study of DNA Methylation of Bladder Cancer Biomarkers in the Urine of Patients with Neurogenic Lower Urinary Tract Dysfunction

**DOI:** 10.3390/biology12081126

**Published:** 2023-08-12

**Authors:** Periklis Koukourikis, Maria Papaioannou, Petros Georgopoulos, Ioannis Apostolidis, Stavroula Pervana, Apostolos Apostolidis

**Affiliations:** 12nd Department of Urology, Aristotle University of Thessaloniki, General Hospital ‘Papageorgiou’, 56403 Thessaloniki, Greece; periskouk@gmail.com (P.K.); petgeo@rm.dk (P.G.); apo.giannis@hotmail.com (I.A.); 2Department of Biological Chemistry, Medical School, Aristotle University of Thessaloniki, 54124 Thessaloniki, Greece; mpapaioannou@auth.gr; 3Pelvic Floor Unit, Department of Urology, Aarhus University Hospital, 8200 Aarhus, Denmark; 4Department of Pathology, General Hospital Papageorgiou, 56429 Thessaloniki, Greece; st.pervana@gmail.com

**Keywords:** DNA methylation, neurogenic lower urinary tract dysfunction, neurogenic bladder, urinary bladder neoplasms, urine biomarker

## Abstract

**Simple Summary:**

As patients with neurogenic bladders are at higher risk for advanced-stage bladder cancer, it is important to develop non-invasive screening methods for bladder cancer other than cystoscopy in this patient population. We conducted a study to explore changes in five bladder cancer-associated genes in the urine of such patients in comparison with healthy controls, and found that changes in certain tumor-suppressing genes associated with bladder cancer were significantly more frequent in the neurogenic group. Apart from a neurogenic bladder, male gender was another risk factor for such gene changes. Bladder biopsies taken from these patients revealed a high percentage of chronic inflammation. We need larger long-term studies to establish the value of this non-invasive method in the screening and diagnosis of bladder cancer, but also to explore possible associations between gene changes in the urine, chronic bladder inflammation and bladder cancer in the neurogenic population.

**Abstract:**

**Background:** Bladder cancer (BCa) in patients suffering from neurogenic lower urinary tract dysfunction (NLUTD) is a significant concern due to its advanced stage at diagnosis and high mortality rate. Currently, there is a scarcity of specific guidelines for BCa screening in these patients. The development of urine biomarkers for BCa seems to be an attractive non-invasive method of screening or risk stratification in this patient population. DNA methylation is an epigenetic modification, resulting in the transcriptional silencing of tumor suppression genes, that is frequently detected in the urine of BCa patients. **Objectives:** We aimed to investigate DNA hypermethylation in five gene promoters, previously associated with BCa, in the urine of NLUTD patients, and in comparison with healthy controls. **Design, setting and participants:** This was a prospective case–control study that recruited neurourology outpatients from a public teaching hospital who had suffered from NLUTD for at least 5 years. They all underwent cystoscopy combined with biopsy for BCa screening following written informed consent. DNA was extracted and DNA methylation was assessed for the RASSF1, RARβ, DAPK, TERT and APC gene promoters via quantitative methylation-specific PCR in urine specimens from the patients and controls. **Results:** Forty-one patients of mixed NLUTD etiology and 35 controls were enrolled. DNA was detected in 36 patients’ urine specimens and in those of 22 controls. In the urine specimens, DNA was hypermethylated in at least one of five gene promoters in 17/36 patients and in 3/22 controls (47.22% vs. 13.64%, respectively, *p* = 0.009). RASSF1 was hypermethylated in 10/17 (58.82%) specimens with detected methylation, APC in 7/17 (41.18%), DAPK in 4/17 (23.53%), RAR-β2 in 3/17 (17.56%) and TERT in none. According to a multivariate logistic regression analysis, NLUTD and male gender were significantly associated with hypermethylation (OR = 7.43, *p* = 0.007 and OR = 4.21; *p* = 0.04, respectively). In the tissue specimens, histology revealed TaLG BCa in two patients and urothelial squamous metaplasia in five patients. Chronic bladder inflammation was present in 35/41 bladder biopsies. **Conclusions:** DNA hypermethylation in a panel of five BCa-associated genes in the urine was significantly more frequent in NLUTD patients than in the controls. Our results warrant further evaluation in longitudinal studies assessing the clinical implications and possible associations between DNA hypermethylation, chronic inflammation and BCa in the NLUTD population.

## 1. Introduction

Bladder cancer (BCa) ranks as the tenth most prevalent cancer globally and is the second most common cancer affecting the urinary system. The likelihood of developing BCa increases with age, and the median age of diagnosis in the general population typically ranges between 60 and 70 years. BCa is more prevalent in men than in women, with their incidence and mortality rates approximately four times higher compared to women worldwide [1]. 

BCa arises from a combination of genetic abnormalities, epigenetic alterations and external risk factors. The primary risk factor is tobacco smoking, followed by occupational exposure to aromatic amines and polycyclic aromatic hydrocarbons. Other risk factors include use of cyclophosphamide and pioglitazone, analgesic abuse, pelvic radiation and infection with parasitic schistosomiasis in areas where it is endemic [2]. Histologically, approximately 90% of BCa cases are classified as transitional cell carcinomas, while squamous cell carcinomas and adenocarcinomas account for about 5% and 2% of cases, respectively [3]. BCa comprises a wide spectrum of disease, non-muscle-invasive cancer (NMIBC; Tis, Ta, T1) can be found in 70–80% of cases, another 10–30% constitute muscle-invasive BCa (MIBC; T_2_–T_4_) and approximately 5% of newly diagnosed BCa patients present with regional or distant metastasis. NMIBC can be effectively treated using transurethral resection of the bladder tumor (TURBT), often in combination with intravesical instillations of chemotherapy or immunotherapy. In MIBC, neoadjuvant therapy followed by radical cystectomy is the treatment of choice. Its outcomes depend on the disease stage, grade and pathological characteristics. Generally, NMIBC has a 5-year progression-free survival rate between 54% and 93%, while the 5-year overall survival rate is 36–48% in MIBC and drops to 5–36% in metastatic disease [4].

BCa is routinely diagnosed via cystoscopy combined with urine cytology in cases of hematuria or lower urinary tract symptoms [5]. However, both methods have several disadvantages. Cystoscopy is an invasive, costly procedure that involves a risk of complications, mainly urinary tract infection and hematuria, thus affecting patient compliance, while its sensitivity rates are 85–90% [6]. Urine cytology, on the other, has inadequate sensitivity for diagnosing low-grade tumors (16%), though different pathologists may variably interpret its findings [7]. Over the past few decades, the utilization of urinary liquid biopsy in BCa has gained increasing interest as it represents a noninvasive sampling method of tumor components released into the urine. A variety of urine tumor biomarkers has been developed for the detection, prognosis and surveillance of BCa, including DNA methylation, gene mutations, protein-based assays, extracellular vesicles, non-coding RNAs and mRNA signatures [8]. Numerous urine biomarkers such as nuclear matrix protein 22 NMP22 (Matritech, Newton, MA, USA), bladder tumor antigen BTA Stat and TRAK tests (Bard Diagnostic Sciences, Redmond, WA, USA), UroVysion (Vysis-Abbot Laboratories, Downers Grove, IL, USA) and ImmunoCyt/uCyt+ (DiagnoCure Inc., Saint-Foy, QC, Canada) have received European Medicines Agency (EMA) and Food and Drug Administration (FDA) approval for BCa diagnosis and surveillance. However, none of these biomarkers have been included in international guidelines, and their use in every day clinical practice remains limited [9]. 

DNA methylation is an epigenetic modification which occurs in CpG dinucleotides throughout the genome, modulating gene (promoter and gene body imprinting) expression, genome stability/integrity and repetitive element repression [10]. Moreover, genome-wide hypomethylation and site-specific hypermethylation, mainly targeting CpG islands in gene promoter regions, characterize cancer cells and play a pivotal role in the early stages of carcinogenesis [11]. Aberrant DNA methylation is commonly observed in BCa, with hypermethylation detected in approximately 50–90% of cases. The application of novel techniques allows for the recognition of DNA hypermethylation in body fluids with high sensitivity and specificity, and several studies support the use of DNA methylation in urine for the early detection of BCa [12,13,14,15,16]. A plethora of genes have been studied in urine samples as possible biomarkers for the initial diagnosis and surveillance of BCa, such as VIM, TWIST, ZNF154, CDH1, APC, RASSF1A, RARβ, GSTP1, DAPK, BLC2, p16 and HOX9, with a sensitivity of ≥90% when used in combination [17,18].

The literature on the study of BCa in patients with neurogenic lower urinary tract dysfunction (NLUTD) extends over more than five decades. Early studies showned incidence rates of up to 10% in spinal-cord-injured (SCI) patients. By contrast, most recent reviews conclude that the overall risk of developing BCa in NLUTD patients is similar to that in the general population [19,20,21]. Despite variations in the reported incidence of BCa among patients with NLUTD, there is consensus for a significantly increased risk of BCa-related mortality, which is 6.7 times higher compared to that in the general population [22]. In NLUTD patients, BCa is diagnosed at an earlier age and more advanced stage, it tends to exhibit more aggressive behavior and it has a high proportion (up to 33.5%) of squamous cell carcinomas [23,24]. Apart from the conventional risk factors observed in the general population, the use of indwelling or intermittent bladder catheters, the presence of bladder stones and recurrent urinary tract infections may be related to the development of chronic inflammation in NLUTD patients, leading to urothelial carcinogenesis [25,26,27]. 

Consequently, there is a requirement to establish an effective diagnostic approach to this uncommon condition in order to enhance its prognosis. To date, guidelines for BCa screening specifically tailored to this population are sparse; expert opinions recommend annual cystoscopy along with wash cytology, with or without biopsy. Conversely, others propose reserving cystoscopy for cases with symptomatic indications, such as gross hematuria, owing to the relatively low incidence of BCa, the high costs associated with screening, and the potential morbidity linked to cystoscopy [28,29,30,31]. The exploration of urine biomarkers appears to be a promising tool for non-invasive screening and early diagnosis of BCa in this specific patient population.

The aim of this study was to investigate the DNA hypermethylation of a panel of five gene promoters known to be associated with BCa (Ras-Association domain Family member 1 (RASSF1), Retinoid Acid Receptor beta (RARβ), Death-Associated Protein Kinase (DAPK), Telomerase Reverse Transcriptase (TERT) and Adenomatous Polyposis Coli (APC)) in the urine of patients suffering from NLUTD and in normal controls, and to evaluate the usefulness of this panel in BCa screening in the NLUTD population. These genes are known to be separately associated with BCa, and the diagnostic potential of this specific panel of genes in BCa has been previously assessed in a non-neurogenic Greek population by our team with encouraging results [32].

## 2. Materials and Methods

### 2.1. Study Design and Participant Enrollment

This was a prospective, case–control, study conducted at the outpatient Neurourology clinic of the Urology department of a public teaching hospital. The present study was conducted in accordance with the ethical standards of the 1975 Declaration of Helsinki, as revised in 2008, and all participants signed written informed consent before their recruitment in the study. The study protocol was approved by the local university BioEthics Committee and registered in the International Traditional Medicine Clinical Trial Registry (ISRCTN37551129). Consecutive patients suffering from NLUTD for at least 5 years, scheduled for cystoscopy with cold cup biopsy and wash cytology for BCa screening were asked to participate in this study. Exclusion criteria included being aged under 18 years, having symptoms and signs of an active urinary tract infection (UTI), and having had a prior diagnosis of and treatment for BCa or another malignancy of the urinary tract. Patients’ demographics, and data on the NLUTD etiology and duration, the method of voiding, catheter usage duration, history of hematuria, recurrent urinary tract infections and bladder stones, were collected. Thirty-five volunteers (patients’ attendants and hospital staff members) without lower urinary tract symptoms or urinary tract diseases were recruited as controls.

### 2.2. Urine Sample Collection and Processing

Fifty milliliters of urine were collected before cystoscopy from each patient in a sterile urine container via spontaneous voiding, clean intermittent catheterization (CIC) or indwelling catheterization (IDC) depending on their voiding habits. Fifty milliliters of free-voiding midstream urine was collected from each control. In all urine samples, the content of the urine preservative single dose (Cat#18124, NORGEN BIOTEK CORP., Thorold, ON, Canada), which stabilizes DNA, RNA and proteins at room temperature for over two years without the need for immediate freezing of the samples, was added. The samples were processed, in a blinded fashion, at the Laboratory of Biological Chemistry of the School of Medicine of the Aristotle University of Thessaloniki. DNA was extracted from urine sediments using the Cells and Tissue DNA Isolation Kit (Norgen Biotek Corp.) and modified with sodium bisulfite using the EZ DNA Methylation-Gold Kit (Zymo Research, Irvine, CA, USA), according to the manufacturer’s instructions. Methylation-specific quantitative PCR (MSP-qPCR) was performed to detect bisulfite-induced changes in the promoters of the following genes: RASSF1, RARβ, DAPK, TERT and APC. MSP-qPCR was performed using Luna Universal Probe qPCR Master Mix (New England Biolabs, Ipswich, MA, USA), as per the manufacturer’s instructions, on an Applied Biosystems StepOnePlus Real-Time PCR System (Thermo Fisher Scientific, Inc., Waltham, MA, USA). Each reaction used 30 ng of modified DNA, and the cycling conditions consisted of an initial step at 95 °C for 1 min, followed by 40 cycles of 95 °C for 30 s and 60 °C for 30 s. The primers and probe were designed to specifically amplify the bisulfite-converted promoters of the genes of interest and their sequences. Positive and negative controls were used and the methylation status of the genes was determined using StepOne™ and StepOnePlus™ Software v2.0. To quantify and compare the amplification products, the Cq data of the target genes were normalized to those of the internal housekeeping gene, actin beta. Standard 100% methylated control human DNA and 100% non-methylated control human DNA (EpiTect PCR Control DNA set, Qiagen, Germany) were used. MSP-qPCR was conducted in triplicate for each sample. The panel of the five genes was recorded as positive when it had at least one hypermethylated promoter.

### 2.3. Cytology

Bladder wash cytology was performed during cystoscopy and the samples were sent directly to the Cytology Laboratory of the hospital. The cytopathologists were blinded to the patients’ cystoscopy findings. Cytology was considered positive when suspicious or malignant cells were detected in the bladder washing samples.

### 2.4. Cystoscopy

Rigid cystoscopy combined with mapping (sidewalls, posterior wall, dome) cold-cup biopsies or biopsies from suspicious sites was performed as an outpatient procedure under local anesthesia by a single, board-certified urologist. The results were reported as normal, suspicious or positive for a bladder tumor. Patients with a bladder tumor or positive cold-cup biopsy were scheduled for transurethral bladder tumor resection (TUR-BT under spinal or general anesthesia.

### 2.5. Bladder Biopsies and Histopathology

Bladder biopsies were processed in the hospital’s pathology laboratory by a single experienced uro-pathologist. The results were reported according to the WHO 2004/2016 histological classification for urothelial carcinomas and flat lesions [33].

### 2.6. Statistical Analysis

Outcomes. The primary outcome of this study was a comparison of the percentages of panel hypermethylation between the NLUTD group and the control group. The secondary outcomes were the diagnostic performance of the panel for BCa diagnosis, and the associations of panel hypermethylation with the demographics and clinical characteristics of the NLUTD group.

Sample size calculation. This is a pilot study and a target of 30 participants was set for each group. Expecting 20% of urine samples without DNA detection, the final target of the participants was at least 35 for each group [34].

Statistics. Descriptive statistics were estimated for each variable in each group. The frequency and relative frequency and the median and interquartile range (IQR) were reported for the categorical and continuous variables, respectively. The two groups were compared using the *t* test for continuous variables or the chi-square test for categorical variables. Univariate and multivariate logistic regression models (factors with a *p*-values > 0.2 were included in the multivariable model) were applied to identify the predictors of hypermethylation. For all statistical analyses, a two-sided *p*-value < 0.05 was defined as statically significant. All statistical analyses were performed using the statistical package R (The R Project for Statistical Computing, Version 3.4.4, for Windows) and R studio (Version 1.1.453 for Windows). 

## 3. Results

Seventy-six subjects were enrolled over the study period, 41 patients (men *n* = 25, women *n* = 16) of mixed NLUTD etiology and 35 healthy controls. The majority of the patients were suffering from spinal cord injury (34.1%) and multiple sclerosis (29.3%) and were emptying their bladders using CIC (82.9%). The subjects’ characteristics are summarized in Table 1. DNA was successfully extracted from 36 of 41 the patients’ and 22 of 35 the controls’ urine samples. DNA was found to be hypermethylated in at least one of five gene promoters in 17/36 patients and in 3/22 controls (47.22% vs. 13.64%, respectively, *p* = 0.009). In the patient group, RASSF1 was hypermethylated in 10/17 (58.82%) urine samples with detected hypermethylation, APC in 7/17 (41.18%) urine samples, DAPK in 4/17 (23.53%) urine samples, RAR-β2 in 3/17 (17.56%) urine samples and TERT in no urine samples (Figure 1). Furthermore, one gene promoter out of five in the panel was hypermethylated in 12/17 (70.59%), and two and three gene promoters were hypermethylated in 3/17 (17.65%) and 2/17 (11.76%), respectively. RASSF1 was the only gene found to be hypermethylated in the control group. In multivariate logistic regression analysis investigating possible correlations between hypermethylation and clinical characteristics, NLUTD and male gender were significantly associated with hypermethylation (OR = 7.43, 95% CI: 1.94–38.00; *p* = 0.007 and OR = 4.21, 95% CI: 1.14–18.6; *p* = 0.04, respectively) (Table 2). In the NLUTD group, no statistical association was found between hypermethylation and age, gender, smoking status, NLUTD duration or history of recurrent UTIs.

Cystoscopy was normal in 24 patients; in 16 patients, the results were recorded as suspicious, and a bladder tumor was detected in one patient. The histology results of the cold-cup biopsies and TUR-Bt revealed pTa low-grade urothelial carcinoma in two patients (simple incidence of bladder cancer: 4.88%) and squamous cell metaplasia in five patients. Hypermethylation in urine was detected in both patients with a BCa diagnosis, while cystoscopy was normal in one of them and cytology was negative in both (Table 3). Chronic bladder inflammation was present in 35/41 biopsy samples (mild: *n* = 11, moderate: *n* = 15, severe: *n* = 9). The cytology results were all negative. 

In the NLUTD patients with detected methylation, the statistical analysis showed 100% sensitivity, 55.88% specificity, a 100% negative predictive value and an 11.76% positive predictive value in this panel of genes for BCa diagnosis. The area under the curve (AUC) of the panel was estimated at 0.77 for BCa diagnosis (Figure 2). The diagnostic performance of the cystoscopy in the NLUTD group was 50% sensitivity, a 58.97% specificity, 95.83% negative predictive value and a 5.88% positive predictive value; the sensitivity of wash cytology was 0%.

## 4. Discussion

BCa screening in the NLUTD population remains a matter of debate, and there is lack of high-level evidence and guidelines for a specific screening protocol [35]. Several studies recommend annual screening cystoscopy combined with urine cytology, while some researchers suggest performing random bladder biopsies during the procedure in order to increase the effectiveness of cystoscopy, to identify hidden pathology and evaluate the risk of future BCa [28,29,30,36]. The use of non-invasive urine biomarkers is an attractive screening tool for BCa in the NLUTD population; however, previous reports have underscored the absence of studies in this field [37,38].

To the best of our knowledge, the current study is the first to examine the DNA methylation of BCa biomarkers in NLUTD patients. We found that DNA hypermethylation in a panel of five gene promoters was significantly more frequent in the urine of patients compared to that of normal individuals. According to our multivariate analysis, patients suffering from NLUTD and males have a greater risk for the detection of hypermethylation in the urine. 

In our NLUTD group, the most commonly observed hypermethylated gene promoter was RASSF1. RASSF1 is a well-studied tumor suppressor gene that participates in cell proliferation, regulation of the cell cycle and the induction of cell apoptosis [39]. Methylation-mediated inactivation of the RASSF1 promoter region has been linked to an elevated risk of various cancer types, including small cell lung cancer, breast cancer, gastric cancer, renal cell carcinoma and nasopharyngeal carcinoma, and is considered an excellent biomarker for BCa since it is rarely found in normal tissue [40]. A meta-analysis demonstrated that individuals with the hypermethylated RASSF1 promoter in their urine samples have a significantly increased risk of BCa, with an OR of 19.82 and a 95% CI of 9.25–42.45 [41]. Furthermore, a recent umbrella review provided strong evidence of RASSF1 methylation for BCa risk (OR = 18.46; 95% CI: 12.69–26.85; I2 = 0%) [42].

The promoter of the APC gene exhibited the second highest frequency of hypermethylation among our NLUTD patients. While APC gene mutations have been commonly associated with familial adenomatous polyposis and colon cancer, recent studies suggest a role of APC promoter hypermethylation in the malignant evolution of BCa [43,44]. A recently conducted meta-analysis provided evidence of a significant association between APC promoter hypermethylation and BCa risk, with an OR of 17.01 and a 95% CI: 7.40–39.07 [45].

We also detected DAPK and RARβ promoter hypermethylation at analogous frequencies. Both gene promoters have been linked and suggested as potential biomarkers for BCa in different studies [14,46,47,48]. In particular, for DAPK promoter hypermethylation, a meta-analysis supports its role in bladder tumorigenesis, showing a significant association between the detection of DAPK promoter methylation and increased BCa risk (overall OR = 5.81; 95% CI: 3.83–8.82), irrespective of the sample used for detection (urine, blood, tissue). There was an approximately four-fold increase in the risk for BCa when methylation was detected in urine (OR = 3.89; 95% CI: 2.13–7.09) [49].

Cancer cells have acquired the capability to avoid senescence through mechanisms that maintain telomere length, primarily via telomerase activation. Telomerase upregulation, which has been observed in up to 90% of malignancies, is a result of multiple mechanisms such as TERT promoter mutations, TERT amplifications, TERT structural variants and epigenetic modifications through TERT promoter methylation [50]. Hypermethylation of the TERT hypermethylated oncological region (THOR), described as an alternative mechanism for gene upregulation through promoter hypermethylation that is distinct from the conventional suppression of gene expression, has been recently related to telomerase activation and disease progression in BCa [51,52]. However, in our study, we did not observe any hypermethylation in the promoter region of the TERT gene, despite its association with BCa in previous research [47,48]. Similarly, in our previous study in a non-neurogenic Greek population, TERT showed low percentages of hypermethylation in BCa patients (2.2%) [32]. As variations in DNA methylation have been observed between different human groups regarding macro- and micro-geographical scales [53], our results indicate that DNA hypermethylation of the TERT promoter may not be a reliable biomarker in the NLUTD and Greek BCa populations.

Known risks factors for BCa such as advanced age, male gender, occupational exposure and smoking are still important in the NLUTD population [54]. In theory, these factors could act synergistically with neurogenic abnormalities, increasing the likelihood of developing BCa in affected individuals. Patients with NLUTD should be particularly vigilant about avoiding exposure to occupational carcinogens and adopting smoking cessation strategies to minimize their risk of developing BCa. In our analysis DNA hypermethylation was not found to be associated with age, gender or smoking status in the NLUTD group. A weak association (OR = 4.21; 95% CI: 1.14–18.6, *p* = 0.04) was observed between DNA hypermethylation and male gender in the multivariate analysis of the entire cohort. This could be attributed to the limited representation of females in the control group. Larger studies with equal numbers of male and female participants would be required for more robust conclusions.

Regarding BCa diagnosis, our panel of genes showed a high sensitivity and negative predictive value of 100%, but with low specificity of 55.88% in the NLUTD group with detected methylation. If our panel was used as a screening tool before cystoscopy, 19 patients could avoid the procedure. Furthermore, in our cohort, the conventional screening tools performed poorly, demonstrating low sensitivity: 50% and 0% for cystoscopy and cytology, respectively. Such results are in line with recent reviews; in a meta-analysis by Gui-Zhong et al., cystoscopy was found to have 64% sensitivity (95% CI: 49.3–76.5%) and urine cytology 36.3% sensitivity (95% CI: 21.5–54.3%) for both the screening and diagnosis of BCa in NLUTD patients [55]. Alimi et al. reported 0% sensitivity for cystoscopy and 71% for urine cytology, and a specificity range of 62–90% for cystoscopy and 92–97% for urine cytology when screening for BCa [56]. Notably, in our cohort, one patient with BCa had normal cystoscopy and negative cytology and would therefore have been missed if we had used only these diagnostic tools without a bladder biopsy.

In our biopsy reports, chronic inflammation in the bladder wall was a prevalent finding in 35 out of 41 cases. Chronic inflammation has been strongly correlated with approximately 20% of all human cancers [57]. In the gastrointestinal tract, this association has been studied extensively. Chronic inflammatory conditions such as inflammatory bowel disease, Helicobacter pylori infection, Barrett’s esophagus, and chronic pancreatitis predispose benign epithelial cells to malignant transformation [58]. 

In the bladder, the chronic inflammation caused by urinary schistosomiasis has been connected with the development of BCa [59]. Furthermore, Schistosoma-associated BCa exhibits analogous characteristics with the BCa found in NLUTD patients, such as the high percentage of squamous cell carcinoma, the advanced stage of the disease and the earlier age at diagnosis. Recently, Cancrini et al. observed similar expression profiles of immunohistochemical biomarkers among neurogenic BCa’s and BCa’s associated with schistosomiasis [60]. In these two conditions, chronic inflammation possibly plays a role in the process of multistage carcinogenesis, which involves tumor initiation, promotion and progression. In an inflammatory microenvironment, exposure to reactive oxygen species/reactive nitrogen species or pro-inflammatory cytokines like interleukin 6 can have transcriptional effects on DNA methyltransferase 1 protein [61]. This leads to increased DNA methylation of tumor suppressor genes. The downregulation of tumor suppressor genes through inflammation-induced promoter DNA methylation can serve as the initial step of carcinogenesis, referred to as “initiation” [62]. 

Most of our study tested genes that have been previously investigated in urinary schistosomiasis and Schistosoma-associated BCa. The hypermethylation of RASSF1A and TIMP3 has been detected in the urine of patients with urogenital schistosomiasis, indicating their potential as biomarkers [63]. Additionally, studies have found DNA hypermethylation of RARbeta2 and APC genes to be potential urine biomarkers for BCa, with higher methylation rates observed in Schistosoma-associated cases [64]. Furthermore, differential methylation of the CDH1, DAPK1, CDKN2A, MGMT, ICDKN2B, FHIT, APC, RASSF1, GSTP1, RARB, and TP73 genes has been observed in BCa specimens, with Schistosoma-associated cases exhibiting a higher number of differentially methylated genes compared to non-associated cases [65]. 

According to these data, the high frequency of hypermethylation in our NLUTD group could be explained provided that DNA hypermethylation exists in individuals even before BCa becomes clinically evident, to an extent that it may not be detectable via cystoscopy or biopsy. Longitudinal studies are required to support the proposal that our panel of genes could be used as an initial screening test for BCa in NLUTD, and that patients with positive hypermethylation are at higher risk for BCa and in need of closer screening and regular follow-ups with cystoscopy and/or bladder biopsy. 

The limitations of our study are the relatively small sample size, including the low percentages of female controls, the small number of detected BCa cases in our cohort affecting the diagnostic accuracy of our panel for BCa in this specific population, and the lack of prospective studies with more BCa cases to validate our results. Furthermore, urinalysis was not concomitantly performed with urine sample collection for DNA methylation to assess the effect of pyuria or bacteriuria in our results. Lastly, the single-center design could have introduced selection bias in our study.

## 5. Conclusions

In this first study evaluating the DNA methylation of BCa-associated genes in the urine of NLUTD patients, DNA hypermethylation in a panel of five genes was significantly more frequent in the urine of NLUTD patients than in that of the controls. In NLUTD patients, our panel of genes could serve as a promising non-invasive biomarker for the risk stratification and diagnosis of BCa. Future longitudinal studies are needed to validate our results and further explore the clinical significance and possible associations of DNA hypermethylation with chronic inflammation and bladder carcinogenesis in this patient population. 

## Figures and Tables

**Figure 1 biology-12-01126-f001:**
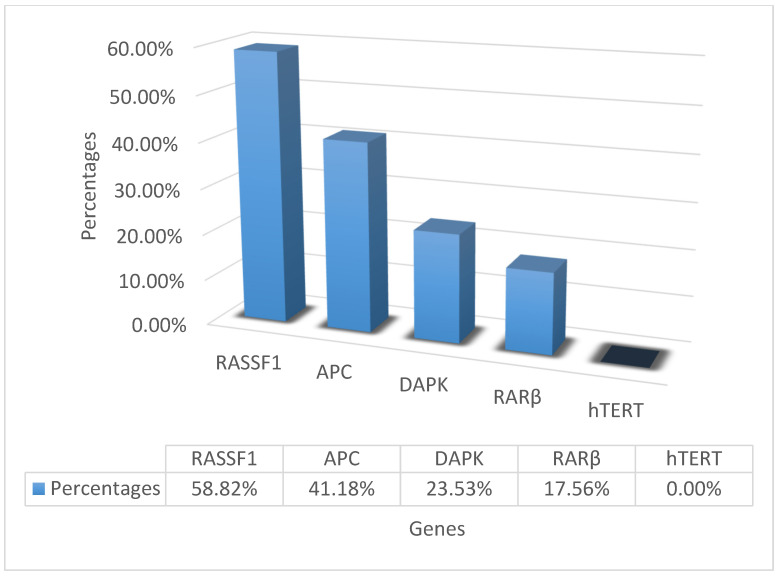
Distribution of hypermethylated genes of panel in NLUTD group.

**Figure 2 biology-12-01126-f002:**
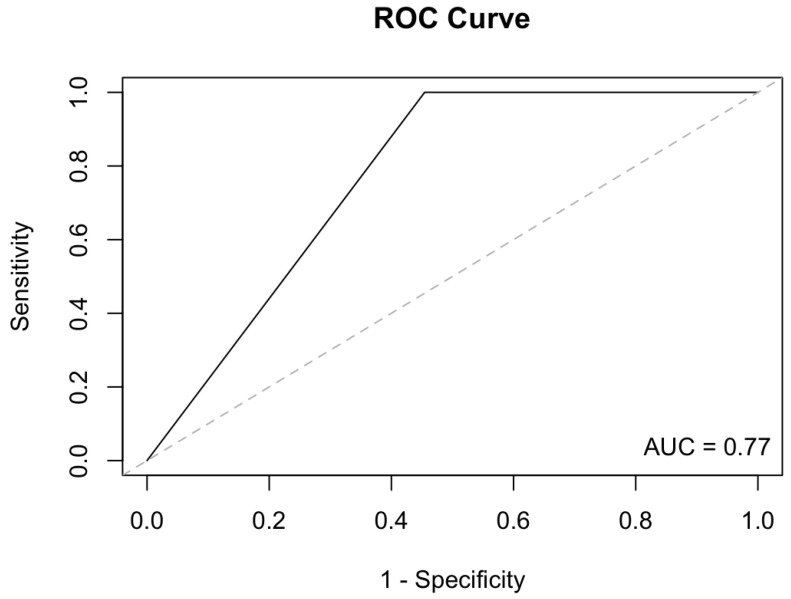
ROC curve and AUC of the gene panel for BCa diagnosis in NLUTD group.

**Table 1 biology-12-01126-t001:** Demographics and clinical characteristics of the study participants.

Variable	NLUTD Group (*n* = 41)	Control Group (*n* = 35)	*p* Value
Age, years (IQR)	49 (17)	62 (31)	0.011
Gender, n (%)			0.072
Male	25 (61.0%)	28 (80.0%)	
Female	16 (39.0%)	7 (20.0%)	
BMI, kg/m^2^, median (IQR)	25.6 (7.3)	27.7 (6.4)	0.12
Smoking, n (%)			0.247
Never	11 (26.8%)	8 (22.9%)	
No	14 (34.1%)	7 (20.0%)	
Yes	16 (39.0%)	20 (57.1%)	
Smoking status, packyears, median (IQR)	16 (30)	20 (46)	0.2
Neurological disease, n (%)			
Spinal cord injury (SCI)	14 (34.1)		
Multiple sclerosis (MS)	12 (29.3)		
Meningomyelocele	5 (12.2)		
Traumatic brain injury	2 (4.9)		
Other	8 (19.5)		
NLUTD duration, years, median (IQR)	10.0 (12.0)		
Method of voiding, n (%)			
CIC	34 (82.9)		
IDC (urethral or suprapubic)	3 (7.3)		
Spontaneous	4 (9.8)		
Catheter usage duration			
(CIC or IDC), years (IQR)	8.0 (8.0)		

**Table 2 biology-12-01126-t002:** Univariable and multivariable logistic regression models evaluating factors associated with hypermethylation. Asterisks stand for statistical significance < 0.05.

Variable	Logistic Regression Models			
	Univariate		Multivariate	
	OR (95% CI)	*p* Value	OR (95% CI)	*p* Value
Group (NLUTD)	5.67 (1.58–27.2)	0.007	7.43 (1.94–38.00)	0.007 *
Age (years)	0.98 (0.95–1.01)	0.24		
Gender (male)	2.91 (0.87–11.7)	0.084	4.21 (1.14–18.6)	0.040 *
BMI(kg/m^2^)	1.00 (0.91–1.09)	0.98		
Packyears	1.0 (0.97–1.02)	0.69		

**Table 3 biology-12-01126-t003:** Demographics, clinical characteristics and diagnostic test results of bladder cancer cases.

Age	Gender	Smoking Status	NLUTD Etiology	NLUTD Duration	Method of Voiding	Cystoscopy	Washing Cytology	Methylation	Histology	Treatment
76	Male	Yes—45 packyears	SCI	17 years	CIC	Bladder tumor	Negative	Positive DAPK	TCC—TaLG	TUR-Bt—surveillance
49	Male	No	SCI	6 years	CIC	Negative	Negative	Positive RARβ	TCC—TaLG	TUR-Bt—surveillance

## Data Availability

Data can become available by the authors upon reasonable request.

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
