# Peer review of "A Study of DNA Methylation of Bladder Cancer Biomarkers in the Urine of Patients with Neurogenic Lower Urinary Tract Dysfunction"

_biology, 2023, doi:10.3390/biology12081126_

Round 1

Reviewer 1 Report

The document is well presented and well structured. The document contains a good number of references, but most of them (94%) are not recent publications (within the last 5 years). It should be noted that it contains a small number of self-citations. The results of the manuscript are reproducible according to the information in the methods section. It is suggested that the discussions be expanded to focus on the metabolomic component. The conclusions are consistent with the evidence and the arguments made in the paper are supported by the citations listed.

Overall, the article stands out for its hypothesis testing; however, the methodology lacked specification of PCR conditions, including the number of cycles, reagent concentration, and operating temperatures. The manuscript is clear and relevant to the field of bladder cancer research, it is well structured, however, most (94%) publications are not recent (within the last 5 years). The experimental design used is appropriate to test the hypothesis. The results of the manuscript are reproducible as indicated in the methods section, except for the PCR conditions, which are not indicated. The figures are generally appropriate. They are easy to interpret and understand and interpret the data appropriately and consistently throughout the manuscript; however, the real-time PCR data are missing when the data are included.

The discussion is too long, considering the number of presented results. The second part of the discussion resembles the review paper, and the authors should focus more on their own results

The conclusions are consistent with the evidence, and the arguments made in the paper are supported by the citations below.

Minor editing of English language required

Author Response

Response

Thank you for the time spent on critically reviewing our manuscript your comments. Based on those, we have made revisions which, we hope, have improved our manuscript.   

Details about PCR conditions were added in the methodology section.

We appreciate your observation about the publication dates of our references. A number of references were replaced by more recent ones and others were added during revision. The percentage of recent publications is now 35%.

As for the discussion section, we summarized and compared our results with previous studies to support our conclusions. We need to point out that the journal has required that our manuscript should be further extended to reach 4000 words, thus we cannot cut down the Discussion section.

Actions

We have added the following text concerning PCR conditions in the Methodology section:

“Methylation-specific quantitative PCR (MSP-qPCR) was performed to detect bisulfite-induced changes at the promoters of the following genes: RASSF1, RARβ, DAPK, hTERT and APC. MSP-qPCR was performed using Luna Universal Probe qPCR Master Mix (New England Biolabs, Massachusetts, USA) as per the manufacturer's instructions on an Applied Biosystems StepOnePlus Real-Time PCR System (Thermo Fisher Scientific, Inc.). Each reaction used 30 ng of modified DNA, and the cycling conditions consisted of an initial step at 95ËšC for 1 minute, followed by 40 cycles of 95ËšC for 30 seconds and 60ËšC for 30 seconds. The primers and probe were designed to specifically amplify the bisul-phite-converted promoter of the gene of interest and their sequences. Positive and negative controls were used and the methylation status of the genes was determined using StepOne™ and StepOnePlus™ Software v2.0. To quantify and compare the amplification products, Cq data of the target genes were normalized to those of the internal housekeep-ing gene, actin beta. Standard 100% methylated control human DNA and 100% non-methylated control human DNA (EpiTect PCR Control DNA set, Qiagen, Germany) were used. MSP-qPCR was conducted in triplicate for each sample.”

Reviewer 2 Report

In the present manuscript entitled “A study of DNA methylation of bladder cancer biomarkers in the urine of patients with neurogenic lower urinary tract dysfunction.” the authors have assessed the clinical value of methylation profile (imprinting of RASSF1, RARβ, DAPK, TERT, APC), regarding diagnosis and prognosis in bladder cancer.

This is a very interesting study, however, there are a number of major issues that authors have to address to be able to publish their results.

Introduction:

1.       Please provide more information on bladder cancer regarding:

-          diagnosis, prognosis and treatment

-          5-years prognosis of different subcategories

-          current clinical need

2.       DNA methylation occurs in CpG dinucleotides throughout the genome, modulating gene (promoter and gene body imprinting) expression, genome stability/integrity and repetitive elements repression. Moreover, genome-wide hypomethylation and site-specific hypermethylation, mainly targeting CpG islands, characterize cancer, including BCa. Please fix the introductory section accordingly.

3.       Please provide more information on liquid biopsy merit in BCa management.

4.       Please add the below references to enhance the introduction:

a.       Sung H, Ferlay J, Siegel RL, Laversanne M, Soerjomataram I, Jemal A, Bray F. Global Cancer Statistics 2020: GLOBOCAN Estimates of Incidence and Mortality Worldwide for 36 Cancers in 185 Countries. CA Cancer J Clin. 2021. doi: 10.3322/caac.21660.

b.       Pilala KM, Papadimitriou MA, Panoutsopoulou K, Barbarigos P, Levis P, Kotronopoulos G, Stravodimos K, Scorilas A, Avgeris M. Epigenetic regulation of MIR145 core promoter controls miR-143/145 cluster in bladder cancer progression and treatment outcome. Mol Ther Nucleic Acids. 2022. doi: 10.1016/j.omtn.2022.10.001.

c.       Thomas T. Bladder EpiCheck for NMIBC. Nat Rev Urol. 2022. doi: 10.1038/s41585-022-00564-7.

Results:

5.       Present the criteria utilized for selecting five specific markers [as acknowledge the existence of other clinically significant epigenetic markers (CDH1, TWIST1, etc) detected in patients' urine].

6.       Visualization of the findings is essential to enhance reader comprehension and facilitate a clearer understanding of the results. In this regard, I suggest to include a Remark diagram of the study, ROC curve (predictive value for BCa diagnosis) and Kaplan-Meier survival curves (you can use KM plotter).

Author Response

General comments

In the present manuscript entitled “A study of DNA methylation of bladder cancer biomarkers in the urine of patients with neurogenic lower urinary tract dysfunction.” the authors have assessed the clinical value of methylation profile (imprinting of RASSF1, RARβ, DAPK, TERT, APC), regarding diagnosis and prognosis in bladder cancer.

This is a very interesting study, however, there are a number of major issues that authors have to address to be able to publish their results.

Suggestion #1

Introduction:

1.       Please provide more information on bladder cancer regarding:

-          diagnosis, prognosis and treatment

-          5-years prognosis of different subcategories

-          current clinical need

Response #1

Thank you for your suggestions. We provided more information on bladder cancer prognosis and treatment in the introduction section. Information about bladder cancer diagnosis and clinical need for the current study are already provided in the introduction section.

Action #1

Two sentences about treatments and prognosis of bladder cancer were added in the introduction section:

“BCa comprises a wide spectrum of disease, non-muscle invasive cancer (NMIBC; Tis, Ta, T1) can be found in 70-80% of the cases, 10-30% constitute muscle invasive BCa (MIBC; T2 – T4) and approximately 5% of newly diagnosed BCa patients present with regional or distant metastasis. NMIBC can be effectively treated using transurethral resection of the bladder tumor (TUR-Bt), often in combination with intravesical instillations of chemotherapy or immunotherapy. In MIBC neoadjuvant therapy followed by radical cystectomy is the treatment of choice. Outcomes depend on disease stage, grade, and pathological characteristics, generally NMIBC has a 5-year progression-free survival rate between 54% and 93% while the 5-year overall survival rate is 36 – 48% in MIBC and drops to 5 – 36% in metastatic disease”

Suggestion #2

    DNA methylation occurs in CpG dinucleotides throughout the genome, modulating gene (promoter and gene body imprinting) expression, genome stability/integrity and repetitive elements repression. Moreover, genome-wide hypomethylation and site-specific hypermethylation, mainly targeting CpG islands, characterize cancer, including BCa. Please fix the introductory section accordingly.

Response #2

Thank you for your suggestion. The introduction section was revised accordingly.

Action #2

Addition of the suggested information about DNA methylation in the Introduction section:

“DNA methylation is an epigenetic modification which occurs in CpG dinucleotides throughout the genome, modulating gene (promoter and gene body imprinting) expression, genome stability/integrity and repetitive elements repression. Moreover, genome-wide hypomethylation and site-specific hypermethylation, mainly targeting CpG islands in gene promoter regions, characterize cancer cells and play a pivotal role in the early stages of carcinogenesis.”

Suggestion #3

Please provide more information on liquid biopsy merit in BCa management.

Response #3

Thank you for your suggestion. We added information on liquid biopsy in the Introduction section.

Action #3

Addition of more information about liquid biopsy in BCa in the Introduction section:

“Over the past few decades, the utilization of urinary liquid biopsy in BCa has gained increasing interest as represents a noninvasive sampling method of tumor components released into the urine. A variety of urine tumor biomarkers has been developed for the detection, prognosis and surveillance of BCa including DNA methylation, gene mutations, protein-based assays, extracellular vesicles, non-coding RNAs and mRNA signatures (8). Numerous urine biomarkers such as nuclear matrix protein 22 NMP22 Matritech, Newton, Massachusetts), bladder tumor antigen BTA Stat and TRAK tests (Bard Diagnostic Sciences, Redmond, Washington), UroVysion (Vysis-Abbot Laboratories, Downers Grove, Illinois) and ImmunoCyt/uCyt+ (DiagnoCure Inc., Saint-Foy, Canada)  have received European Medicines Agency (EMA) and Food and Drug Administration (FDA) approval for BCa diagnosis and surveillance. However, none of these biomarkers have been included in international guidelines and their use in every day clinical practice remains limited (9).” 

Suggestion #4

Please add the below references to enhance the introduction:

a.       Sung H, Ferlay J, Siegel RL, Laversanne M, Soerjomataram I, Jemal A, Bray F. Global Cancer Statistics 2020: GLOBOCAN Estimates of Incidence and Mortality Worldwide for 36 Cancers in 185 Countries. CA Cancer J Clin. 2021. doi: 10.3322/caac.21660.

b.       Pilala KM, Papadimitriou MA, Panoutsopoulou K, Barbarigos P, Levis P, Kotronopoulos G, Stravodimos K, Scorilas A, Avgeris M. Epigenetic regulation of MIR145 core promoter controls miR-143/145 cluster in bladder cancer progression and treatment outcome. Mol Ther Nucleic Acids. 2022. doi: 10.1016/j.omtn.2022.10.001.

c.       Thomas T. Bladder EpiCheck for NMIBC. Nat Rev Urol. 2022. doi: 10.1038/s41585-022-00564-7.

Response #4

Thank you for your suggestion. The aforementioned references were added in the introduction section.  They now show as references 1, 10 and 18, respectively.

Action#4

Addition of the suggested references.

Suggestion #5

Results:

Present the criteria utilized for selecting five specific markers [as acknowledge the existence of other clinically significant epigenetic markers (CDH1, TWIST1, etc) detected in patients' urine].

Response #5

Thank you for your comment. The diagnostic accuracy for bladder cancer of this specific panel of genes has been previously assessed in non-neurogenic Greek population by our team and the results were encouraging (reference Georgopoulos et al.). As DNA methylation has been described to be an early event on bladder carcinogenesis, we tested the same panel for screening in the NLUTD population, who appear to be at increased risk for bladder cancer. As we already mentioned in the Introduction section, these genes are known to be separately associated with BCa, and we provided more analytical information of these associations in Discussion.

Action #5

Addition of a sentence about the selection of the genes of our panel in the Introduction section:

“These genes are known to be separately associated with BCa and the diagnostic potential of this specific panel of genes in BCa has been previously assessed in non-neurogenic Greek population by our team with encouraging results”

Suggestion #6

Visualization of the findings is essential to enhance reader comprehension and facilitate a clearer understanding of the results. In this regard, I suggest to include a Remark diagram of the study, ROC curve (predictive value for BCa diagnosis) and Kaplan-Meier survival curves (you can use KM plotter).

Response#6

Thank you for your suggestion.

A figure of ROC curve for BCa diagnosis was added in the manuscript and a sentence was added in the Results section.

Remark diagrams are mainly used to outline the key components that should be included in the reporting of tumor marker prognostic studies. Furthermore, the Kaplan-Meier method and the survival curves are commonly used to estimate survival probabilities over time. The current study was a case – control study which assessed the usefulness of a panel of genes in BCa screening in the NLUTD population at a single time-point and prognostic or survival data were not obtained to include a Remark diagram or Kaplan-Meier survival curves.

Action#6

Addition of “Figure 2. ROC curve and AUC of the gene panel for BCa diagnosis in NLUTD group.” and addition of a sentence in the Results section: “The area under the curve (AUC) of the panel was estimated at 0.77 for BCa diagnosis.”

Reviewer 3 Report

A pilot study examining non-invasive methods of screening and diagnosing BCa in patients with neurogenic abnormalities is appreciated. The study, in its limitation, identifies a panel of five genes (RASSF1, RARβ, DAPK, APC, and hTERT) as BCa biomarkers, in the samples collected from patients with neurogenic abnormalities, which rely on the hypermethylation of these genes. The sensitivity of detection is 100% which is amazing. Nevertheless, my concerns are:

1-    What was the basis of selection of these specific genes (RASSF1, RARβ, DAPK, APC and hTERT) as BCa biomarkers? What was the approach for consideration? Do these panels of genes detect LUTD when the cause is neurogenic only?

2-    The text mentions risk factors such as tobacco smoking and occupational exposure to AA, PAH, etc. are associated with occurrence of BCa. It could be apt to the present study if these risk factors were considered and compared as well. Can you explain whether these risk factors magnify the occurrence of BCa in patients with neurogenic abnormalities? What do you expect the outcome of DNA methylation if these risk factors were included as well?

3-    Spinal cord injury or other neurogenic abnormalities lead to urinary tract infections (UTI) due to microorganisms’ interference. How do you see the outcome on the expression of panels of genes considering the UTI involving microorganisms? Moreover, it is discussed male have greater risk for detection of hypermethylation in urine. Why? Wouldn't you have liked to have had more samples from both groups (males and females) tested before drawing any conclusions?

4-    hTERT showed 0% hypermethylation activity in the study, yet included in the panel of genes as BCa biomarkers. Explain

No issues with English Language!

Author Response

General comments

A pilot study examining non-invasive methods of screening and diagnosing BCa in patients with neurogenic abnormalities is appreciated. The study, in its limitation, identifies a panel of five genes (RASSF1, RARβ, DAPK, APC, and hTERT) as BCa biomarkers, in the samples collected from patients with neurogenic abnormalities, which rely on the hypermethylation of these genes. The sensitivity of detection is 100% which is amazing. Nevertheless, my concerns are:

Suggestion #1

What was the basis of selection of these specific genes (RASSF1, RARβ, DAPK, APC and hTERT) as BCa biomarkers? What was the approach for consideration? Do these panels of genes detect LUTD when the cause is neurogenic only?

Response #1

Thank you for your comment. The diagnostic accuracy for bladder cancer of this specific panel of genes has been previously assessed in non-neurogenic Greek population by our team and the results were encouraging (reference Georgopoulos et al.). As DNA methylation has been described to be an early event in carcinogenesis we tested the same panel for screening in the NLUTD population, who appear to be at increased risk for advanced-stage bladder cancer. As we already mentioned in the Introduction section, these genes are known to be separately associated with bladder Ca, and we provided more analytical information of these associations in Discussion. Thus, the aim of this study was to evaluate the usefulness of this panel in bladder screening in the NLUTD patients. The panel could be used for bladder cancer diagnosis and risk stratification for bladder cancer in this population. There is currently no published literature to suggest an association between LUTD of either neurogenic or non-neurogenic origin and this panel of genes.

Action #1

Addition of a sentence about the selection of the genes of our panel in the Introduction section:

“These genes are known to be separately associated with BCa and the diagnostic potential of this specific panel of genes in BCa has been previously assessed in non-neurogenic Greek population by our team with encouraging results”

Suggestion #2

The text mentions risk factors such as tobacco smoking and occupational exposure to AA, PAH, etc. are associated with occurrence of BCa. It could be apt to the present study if these risk factors were considered and compared as well. Can you explain whether these risk factors magnify the occurrence of BCa in patients with neurogenic abnormalities? What do you expect the outcome of DNA methylation if these risk factors were included as well?

Response #2

Thank you for your observation. Data for risk factors as tobacco smoking and occupational exposure were obtained from our study subjects. None of the participants had a history of occupational exposure to AA or PAH. Tobacco smoking data are presented in Table 1 of the manuscript. As we described in Results section, in the NLUTD group we examined the risk factor of tobacco smoking and no statistical association was found between hypermethylation and smoking status.

In theory the risk factors of smoking and occupational exposure could act synergistically with neurogenic abnormalities, increasing the likelihood of developing BCa in affected individuals. Patients with NLUTD should be particularly vigilant about avoiding exposure to occupational carcinogens and adopting smoking cessation strategies to minimize their risk of developing BCa.

Action #2

Addition of a paragraph regarding known risk factors for BCa and DNA methylation in Discussion section:

“Known risks factors for BCa such as advanced age, male gender occupational exposure and smoking are still important in NLUTD population (55). In theory, these factors could act synergistically with neurogenic abnormalities, increasing the likelihood of developing BCa in affected individuals. Patients with NLUTD should be particularly vigilant about avoiding exposure to occupational carcinogens and adopting smoking cessation strategies to minimize their risk of developing BCa. In our analysis DNA hypermethylation was not found to be associated with age, gender or smoking status in the NLUTD group.”

Suggestion #3

Spinal cord injury or other neurogenic abnormalities lead to urinary tract infections (UTI) due to microorganisms’ interference. How do you see the outcome on the expression of panels of genes considering the UTI involving microorganisms? Moreover, it is discussed male have greater risk for detection of hypermethylation in urine. Why? Wouldn't you have liked to have had more samples from both groups (males and females) tested before drawing any conclusions?

Response #3

Thank you for your interesting observation. Patients with signs and symptoms of an active UTI were excluded from our study as they cannot undergo cystoscopy with an active infection. Unfortunately, in our study we did not plan to obtain data on pyuria or asymptomatic bacteriuria and a speculation about the association of hypermethylation of our panel of genes with these factors could not be made.

Regarding the effect of gender, this was a pilot study exploring associations of various risk factors with hypermethylation of our genes’ panel and results can only be considered as indicative. The low percentage of females of the control group could be a reason for this weak association (p=0.04) between hypermethylation and male gender in the entire cohort, while this association was not observed in the NLUTD group when a separate analysis was performed. As the reviewer accurately points out, larger studies with equal numbers of male and female participants would be required for more robust conclusions.

Action #3

Comments about the gender effect in our results and about pyuria and bacteriuria were added in the Discussion section and in the limitations of our study:

“A weak association [OR:4.21 (1.14 - 18.6), p=0.04) was observed between DNA hypermethylation and male gender in the multivariate analysis of the entire cohort. This could be attributed to the limited representation of females in the control group. Larger studies with equal numbers of male and female participants would be required for more robust conclusions”

“Limitations of our study are the relatively small sample size including the low percentages of female controls. Furthermore, urinalysis was not concomitantly performed with urine sample collection for DNA methylation to assess the effect of pyuria or bacteriuria in our results.”

Suggestion #4

hTERT showed 0% hypermethylation activity in the study, yet included in the panel of genes as BCa biomarkers. Explain

Response#4

Thank you for your comment. As we mentioned in the discussion section TERT hypermethylation in a specific region (THOR) leads to high telomerase activity and has been associated with bladder carcinogenesis. For these reasons we included TERT in our panel. The 0% hypermethylation could be explained by the variations in DNA methylation observed between different human groups regarding macro- and micro-geograph­ical scales. Similarly, in our previous study in non-neurogenic Greek population TERT showed low percentages of hypermethylation in BCa patients (2.2%). We revised the paragraph and added more information about TERT methylation

Action#4

Revision of the paragraph about TERT and addition of more information and literature about TERT methylation:

“Cancer cells have acquired the capability to avoid senescence through mechanisms that maintain telomere length, primarily via telomerase activation. Telomerase upregulation, which has been observed in up to 90% of malignancies, is a result of multiple mechanisms such as TERT promoter mutations, TERT amplifications, TERT structural variants and epigenetic modifications through TERT promoter methylation (47). Hypermethylation of TERT hypermethylated oncological region (THOR), described as an alternative mechanism for gene upregulation through promoter hypermethylation that is distinct from the conventional suppression of gene expression, has been recently related with telomerase activation and disease progression in BCa (48,49). However, in our study, we did not observe any hypermethylation in the promoter region of the TERT gene, despite its association with BCa in previous research (43,44). Similarly, in our previous study in non-neurogenic Greek population TERT showed low percentages of hypermethylation in BCa patients (2.2%) (28). As variations in DNA methylation has been observed between different human groups regarding macro- and micro-geographical scales, our results indicate that DNA hypermethylation of the TERT promoter may not be a reliable biomarker in the NLUTD and BCa Greek population.”

Round 2

Reviewer 2 Report

The authors have addressed the proposed suggestions and I believe that the revised manuscript is significantly refined and available for publication.

Reviewer 3 Report

No Comments.

Thanks for the clarifications and responses to the raised questions.......

 English language is good.

Round 3

Reviewer 2 Report

The authors have addressed the proposed suggestions and I believe that the revised manuscript is significantly refined and available for publication.